# Derivation of the Microbial Inactivation Rate Equation from an Algebraic Primary Survival Model Under Constant Conditions

**DOI:** 10.3390/foods14111980

**Published:** 2025-06-03

**Authors:** Si Zhu, Bing Li, Guibing Chen

**Affiliations:** Center for Excellence in Post-Harvest Technologies, North Carolina A&T State University, The North Carolina Research Campus, 500 Laureate Way, Kannapolis, NC 28081, USA; szhu@ncat.edu (S.Z.); bli@aggies.ncat.edu (B.L.)

**Keywords:** process system, state, path, inactivation rate, primary model

## Abstract

A food pasteurization or sterilization process was treated as a system comprising a target microorganism, a food medium, and applied lethal agents (both thermal and nonthermal). So, the state of such a system was defined by the target microorganism’s concentration, the food medium parameters (food composition, pH, and water activity), and the magnitudes of temperature and nonthermal lethal agents. Further, a path was defined as a series of profiles that describe the changes in state factors over time when a food process system changes from its initial state to any momentary state. Using the Weibull model as an example, results showed that, if the microbial inactivation rate depends on path, then there exists an infinite number of rate equations that can result in the same algebraic primary model under constant conditions but, theoretically, only one of them is true. Considering the infinite possibilities, there is no way to find the most suitable or true rate equation. However, the inactivation rate equation can be uniquely derived from the algebraic primary model if the inactivation rate does not depend on path, which was demonstrated to be true by most microbial survival data reported in previous studies.

## 1. Introduction

Raw foods may contain pathogenic microorganisms that cause foodborne diseases, as well as enzymes and spoilage microorganisms that can cause food spoilage. Therefore, these foods need to be pasteurized or sterilized for safe consumption and extended shelf-life. During pasteurization or sterilization of foods, the ratio of the concentration of the target microbial cells or spores at any time, N(t), to their initial concentration, $N_{0}$, is known as the survival ratio, $S\left(t\right)$. A plot of ${log}_{10}S(t)$ versus processing time $\left(t\right)$ is called a survival curve. Under constant processing conditions (e.g., isothermal, isobaric, constant concentration), microbial survival curves in a specific food medium are usually described by a primary mathematical model, which expresses ${log}_{10}S(t)$ as a function of time, and secondary models that describe how the kinetic parameters in the primary model vary as a function of the magnitudes of the lethal agents used (e.g., temperature, pressure, concentration). Numerous investigations have shown that survival curves of various microorganisms, including bacteria, viruses, fungi, and parasites, under constant conditions exhibit various patterns that are described by different mathematical models, including the first-order model [1,2], Weibull model [3,4], log-logistic model [5], modified Gompertz model [6], biphasic first-order model [7], and biphasic Weibull model [8], among others.

In actual food processing, the magnitudes of temperature and/or nonthermal lethal agents used vary with time due to come-up time (CUT) as well as the dynamic nature of heat transfer in foods, for example, temperature at the slowest heating point of canned foods changes throughout the thermal process [9]. Many studies have shown that the accumulated microbial inactivation during CUT was significant and up to several log_10_ reductions depending on specific process conditions [10,11]. Therefore, the microbial inactivation rate equation is needed to calculate the accumulated lethality of the target microorganism under nonconstant, also called dynamic or time-varying, processing conditions to ensure that food processes meet food safety standards.

In several studies, the microbial inactivation rate equations were given directly [12,13,14,15,16]. However, this method is generally challenging mainly for two reasons. First, the inactivation mechanism can be various. This is reflected by various patterns exhibited by microbial survival curves under constant conditions, depending on the specific microorganism, the lethal agents used, and the conditions of the environment (e.g., pH and moisture). A complex pattern of microbial survival curves may consist of a shoulder and a tail, between which there is a linear or nonlinear (concave up or concave down) curve. Second, the pattern of survival curves does not provide directly enough information about the mathematical structure of the inactivation rate equation. On the other hand, under constant conditions, an algebraic function of time or algebraic primary model can usually be obtained to describe survival curves without much difficulty, simply based on the observed pattern of survival curves. Therefore, it is more effective to construct an inactivation rate equation from an algebraic primary model.

For a given algebraic primary model under constant conditions, two different methods have been used to construct the inactivation rate equation. One is assuming that the local slope of a nonconstant survival curve is equal to that of the constant survival curve at the momentary magnitude of the applied lethal agents but at a time that corresponds to the momentary survival ratio [17]. This method has been successfully used in many studies [18,19,20,21,22,23,24]. More details regarding this approach can be found in a recent review [25]. The other is taking the derivative of an algebraic primary model with respect to time as the inactivation rate [26,27,28,29,30,31,32,33,34,35,36]. At present, there is no consensus on which method is more appropriate.

The objective of this study was to come up with a logical and effective way for constructing the inactivation rate equation from an algebraic primary model that describes survival curves under constant conditions using the Weibull model as an example.

## 2. Materials and Methods

### 2.1. Definitions of Food Process System, System State, and Path of State Change

A food pasteurization or sterilization process was treated as a system comprising a target microorganism, food medium, and applied lethal agents (both thermal and nonthermal). The state of such a process system was then defined by the target microorganism’s concentration, food medium parameters (food composition, pH, and water activity), and the magnitudes of temperature (may or may not be at a lethal level) and nonthermal lethal agents, such as high pressure, pulsed-electric field, high-intensity ultrasound, cold plasma, etc., that may be used together with heat. During food processing, only those state factors that may change values are usually of interest, which include the target microorganism’s concentration $\left[N\left(t\right)\right]$, temperature $\left(T\right)$, nonthermal (*NT*) lethal agents, and water activity $\left(a_{w}\right)$ in some cases, for example, low-moisture sterilization processing [37,38]. Except $a_{w}$, other food medium parameters are treated as constants, although the process may cause degradation of some vitamins and phenolic compounds [39] since it does not lead to significant changes in microbial resistance to the applied lethal agents. So, the momentary state of a food pasteurization or sterilization process can be simplified as a vector $ST\left(t\right)=\left[{log}_{10}S\left(t\right),\;T\left(t\right),\;NT\left(t\right),\;a_{w}\left(t\right)\right]$, where $S\left(t\right)$ is the survival ratio that is equal to $N\left(t\right)/N_{0}$, where $N\left(t\right)$ and $N_{0}$ are the concentrations of microbial cells or spores at any time t and time zero, respectively. Among the state factors, $T\left(t\right),\;NT\left(t\right),{\;and\;a}_{w}\left(t\right)$ are independent factors, so they are grouped together and denoted as ${C(t)=[T(t),\;NT(t),\;a}_{w}(t)]$, whereas ${log}_{10}S\left(t\right)$ results from the accumulative effect of C(t) over a period of time. Then, ST(t) can be written as $ST\left(t\right)=\left[{log}_{10}S\left(t\right),C(t)\right]$.

During food processing, the state of a process system changes with time. A path is then defined as a series of profiles that describe the changes in state factors over time when a food process system moves from its initial state [ST(*0*)] to any momentary state [ST(*t*)]. Theoretically, there exist infinite paths for a food process system to reach a given state. The momentary microbial inactivation rate may or may not depend on path.

### 2.2. The Path-Independent Microbial Inactivation Rate

When a target microorganism is lethally treated at a constant ${C=[T,\;NT,\;a}_{w}]$ in a specific food medium, the microbial survival curve can generally be described by an algebraic function of time [Equation (1)].(1)$$\log_{10}S\left(t\right)=f\left(t,\beta_{1}(C),\cdots ,\;{\beta_{i}\left(C\right),\cdots ,\beta }_{j}(C)\right)$$
where $\beta_{1},\cdots ,\beta_{i},\cdots \;\beta_{j}$ denote model parameters that depend on C, and their relationships, i.e., the secondary models, could be established from experimental data using the curve fitting method [40].

If the microbial inactivation rate at any state $ST(t)=[\log_{10}S\left(t\right),C\left(t\right)]$ is solely determined by its current state, and not by the specific path taken to reach this state, then the momentary inactivation rate equals the slope of the survival curve under a constant condition $C=C\left(t\right)$ [Equation (1)] at the point ${[f}^{-1}\left({log}_{10}S(t)\right),\;{log}_{10}S(t)]$, because the current state can be assumed to be reached under the constant condition. The rate equation can then be expressed by Equation (2).(2)$$\frac{d\;\log_{10}S\left(t\right)}{dt}=f^{\prime }\left(t\right){|}_{{t=f}^{-1}\left({log}_{10}S(t)\right)}\;$$
where $f^{-1}$ denotes the inverse function of f.

After the secondary models were established from microbial survival data, Equation (2) was used to numerically calculate microbial survival curves under any condition using a recursive equation expressed by Equation (3) obtained by following a reported procedure [41].(3)$${log}_{10}S_{i}=f\left[f^{-1}\left({log}_{10}S_{i-1}\right)+dt\right]$$
where i is a sequential index, dt a sufficiently small-time interval, and values of model parameters (β) were calculated based on the secondary models at the average C during dt. Starting at $t=0\;and\;{log}_{10}S_{0}=0$, other log_10_ reductions were calculated using Equation (3).

### 2.3. The Path-Dependent Microbial Inactivation Rate

As mentioned above, theoretically, there exist infinite paths to reach a given state. As a result, if the microbial inactivation rate at any state $ST(t)=[\log_{10}S\left(t\right),\;C\left(t\right)]$ depends on path, there are infinite possible microbial inactivation rate equations that can result in Equation (1). For example, assuming the inactivation rate is described as $\frac{d\;\log_{10}S\left(t\right)}{dt}=f^{\prime }\left(t,\beta_{1},\cdots ,\;{\beta_{i},\cdots ,\beta }_{j}\right)$, in which assuming model parameters depend on path in a way $\beta_{i}=\frac{(m+1)\int_{0}^{t}\tau^{m}\cdot \beta_{i}\left[C(\tau )\right]\cdot d\tau }{t^{m+1}}$, where m can be any real number. When C is maintained constant during a food process, for any values of *m*, a simple integration gives $\beta_{i}=\beta_{i}(C)$, and further integration on both sides of the rate equation yields Equation (1).

When an inactivation rate equation was chosen, it was used to calculate microbial survival curves under any processing condition using the Runge–Kutta method [14].

### 2.4. Statistical Analysis

The root-mean-squared error (RMSE) between experimentally measured and calculated values of the microbial log reduction was used to evaluate the performance of inactivation rate equations when predicting the survival curves under dynamic processing conditions. The RMSE value can be described as [24].(4)$$RMSE=\sqrt{\frac{\sum_{i}^{m_{e}}{\left(\log_{10}S_{i,measured}-\log_{10}S_{i,calculated}\right)}^{2}}{m_{e}-p}}$$
where $m_{e}$ stands for the number of experimental data points, and p is the number of parameters to be estimated.

## 3. Results and Discussion

### 3.1. The Relationship Between an Inactivation Rate Equation and an Algebraic Primary Model Under Constant Conditions

Without loss of generality, the Weibull model under thermal processing was taken as an example to show the relationship between an inactivation rate equation and an algebraic primary model under constant conditions. It was assumed that survival curves of a target microorganism in a specific food medium with unchanged water activity under isothermal processing could be described by the Weibull model expressed by Equation (5) [3].(5)$$\log_{10}S\left(t\right)=-b\left(T\right)t^{n}$$
where b is the rate coefficient, which depends on temperature (*T*), whereas n is the power of the Weibull model, which is independent of temperature in most cases [42]. $b\left(T\right)$ is described by Equation (6) [40].(6)$$b\left(T\right)=\ln\left\{1+exp\left[k\left(T-T_{c}\right)\right]\right\}$$
where $T_{c}$ marks the magnitude of temperature around which the inactivation accelerates and k is the slope of the approximately linear $b\left(T\right)$ vs. T relationship when ${T\geq T}_{c}$. According to the model, when ${T\leq T}_{c}$, $b\left(T\right)\approx 0$. More details regarding the application of this secondary model can be found in our previous publications [24,43,44].

There were infinite possible paths (temperature profiles) for a food process system to reach a momentary state $ST\left(t\right)=\left[{log}_{10}S_{m},\;T_{m}\right]$ from its initial state $ST\left(0\right)=\left[0,\;T_{0}\right]$. Figure 1 shows three typical patterns of temperature profiles (paths) and the corresponding survival curves. Under constant temperature conditions with a negligible CUT, survival curves could be approximately described by an algebraic function of time, which was the Weibull model in the present study.

If the microbial inactivation rate does not depend on path, the rate equation, Equation (7), was uniquely derived from Equation (5) using Equation (2).(7)$$\frac{d\;\log_{10}S\left(t\right)}{dt}=-n\cdot {\left\{b\left[T\left(t\right)\right]\right\}}^{\frac{1}{n}}{\cdot \left[-\log_{10}S\left(t\right)\right]}^{\frac{n-1}{n}}$$

On the other hand, Equation (5) could be obtained by integrating Equation (7) at a constant temperature. Therefore, there was a one-to-one relationship between the inactivation rate equation and the primary model. In a previous study, Equation (7) was also obtained by assuming that the local slope of the nonconstant survival curve is equal to that of the constant survival curve at the momentary magnitude of the applied lethal agent but at a time that corresponds to the momentary survival ratio [17]. This assumption is valid if, and only if, the microbial inactivation rate does not depend on path as described in Section 2.2.

If the microbial inactivation rate depends on path, there exists an infinite number of inactivation rate equations due to the existence of an infinite number of paths from the original state at time zero to the state at a given time point. To demonstrate that different rate equations resulted in different calculated microbial survival curves under the same nonconstant heating temperature profile, the following rate equations [Equations (8)–(10)] were constructed from Equation (5). Among them, Equation (9) was obtained by taking the derivative of both sides of Equation (5). It is easy to know that integrating both sides of each of the rate equations resulted in Equation (5) at a constant temperature since *b* is a constant.(8)$$\frac{d\;\log_{10}S\left(t\right)}{dt}=-n\cdot {\left\{\frac{(m+1)\int_{0}^{t}\tau^{m}\cdot b\left[T(\tau )\right]\cdot d\tau }{t^{m+1}}\right\}}^{\frac{1}{n}}{\cdot \left[-\log_{10}S\left(t\right)\right]}^{\frac{n-1}{n}}\;$$(9)$$\frac{d\;\log_{10}S\left(t\right)}{dt}=-n\cdot b\left[T\left(t\right)\right]\cdot t^{n-1}$$(10)$$\frac{d\;\log_{10}S\left(t\right)}{dt}=-n\cdot \frac{\left(m+1\right)\int_{0}^{t}\tau^{m}\cdot b\left[T\left(\tau \right)\right]\cdot d\tau }{t^{m+1}}\cdot t^{n-1}$$

In Equations (8) and (10), m is any real number. There are more possibilities, for example, multiplying the right sides of Equations (8)–(10) by an exponential term $e^{a\cdot \left(dT/dt\right)}$, in which *a* is a positive coefficient, also resulted in Equation (5) under constant temperature, since the exponential term becomes 1.

Under the same temperature profile described by Equation (11), survival curves (Figure 2) were calculated using Equations (7)–(10) [Let m=1  for Equations (8) and (10)] with the same model parameters $k=0.348{\;{}^{\circ}C}^{-1},\;T_{c}=66.34\;{}^{\circ}C,\;and\;n=0.545$ for *Salmonella* enteritidis [45].(11)$$T\left(t\right)=\left\{\begin{array}{c} 25+100t\;\;\;\;\;\;\;\;\;\;\;\;\;\;\;\left(t\leq 0.5\right)\\ 5\sin\left(5t\right)+75\;\;\;\;\;\;\;\left(t>0.5\right) \end{array}\right.$$

As expected, the resulting survival curves were different. As a result, if the inactivation rate depends on path, since there is an infinite number of possible inactivation rate equations that give the same algebraic primary model under constant temperature, the only way to identify the “true” inactivation rate equation is to examine how well each of them predicts survival curves under nonconstant temperature conditions. This is literally an impossible task considering the infinite possibilities.

### 3.2. A Revisit to Previous Studies

#### 3.2.1. The Application of the Path-Independent Inaction Rate Equation

When the microbial inactivation rate does not depend on path, it can be inferred that the local slope of the nonconstant survival curve is equal to that of the constant survival curve at the momentary magnitude of the lethal agent used but at a time that corresponds to the momentary survival ratio, which was also a direct assumption made by Peleg and Penchina [17]. Based on this assumption, the Weibull model [Equation (7)] was used to describe survival curves of various microorganisms in different food media during thermal processing, and all studies found a good agreement between observed and predicted survival curves. For example, the survival of *Salmonella* cells in two sugar-rich media was examined during heating from 65 to 80 °C [18]. In other studies, nonisothermal survival curves of *Bacillus sporothermodurans* IC4 spores and *Bacillus coagulans* spores in soups and nutrients were accurately constructed from their isothermal survival curves using the same method [19,20]. More recently, the method was used for predicting survival of *Escherichia coli* O157:H7 and *Salmonella typhimurium* during microwave pasteurization of apple juice [23]. The method also showed its validity for nonthermal processing, such as high hydrostatic pressure processing [21], pulsed electrical field [46], plasma [47], and thermo-ultrasonic treatment [22]. Apart from the Weibull model, the assumption was shown to be applicable for other microbial survival models, including log-normal distribution [48], Gompertz model [49], shifted logistic model [50,51], and several other not frequently used models [17,52,53].

In several previous studies, the microbial inactivation rate equations were developed directly and were listed in Table 1. By integrating these rate equations under constant conditions, the survival models were obtained and listed in the table. It was found that the same rate equations could be obtained from the survival models using Equation (2). This proved that the microbial inactivation rate in these studies did not depend on path.

#### 3.2.2. The Application of the Path-Dependent Inaction Rate Equations

In some previous studies, inactivation rate equations were derived by taking the derivative of both sides of an algebraic primary model under constant conditions. This method inherently assumed path-dependence of inactivation rates, since only the rate equation derived using Equation (2) did not depend on path. For the Weibull model, Equation (9) was obtained in this way and used to calculate microbial survival curves under nonconstant conditions in some studies [26,27,28,29,30,31,32,33]. The same method was also used for the Gompertz model [33,34,35,55,56,57,58]. Although these studies found that the method provided reasonably good predictions, it could not rule out other possible rate equations that might outperform this one since, statistically, there was a near-zero probability to obtain the right one by randomly choosing one from infinite possibilities.

#### 3.2.3. Case Studies

Four sets of previously reported survival data were revisited to compare the performance of path-independent [Equation (7)] and path-dependent [Equation (9)] inactivation rate equations in predicting survival curves under nonconstant processing conditions. In Case I, Esteban, Huertas, Fernandez and Palop [32] studied the heating medium characteristics (pH and food matrix) on the thermal inactivation of *Bacillus sporothermodurans* IC4 spores when subjected to isothermal and non-isothermal heating and cooling treatments. Other cases included isothermal and non-isothermal treatments of *Geobacillus stearothermophilus* T26 in distilled water [29] (Case II), *Listeria monocytogenes* in ground beef [33] (Case III), and *Staphylococcus aureus* and *Salmonella senftenberg* in peptone water [59] (Case IV).

For each of the cases, survival curves under isothermal treatment were fitted with Equation (12), which is another expression of the Weibull model, to determine the model parameters.(12)$$\log_{10}S=-{\left(\frac{t}{\delta \left(T\right)}\right)}^{n}$$
where $\delta \left(T\right)$ is the rate coefficient, and its relationship with b(T) in Equation (5) is described by Equation (13).(13)$$b(T)={\left(\frac{1}{\delta (T)}\right)}^{n}$$

The temperature dependence of $\delta \left(T\right)$ is given by Equation (14) [29].(14)$$\delta \left(T\right)=\delta_{T_{ref}}\times 10^{\frac{T_{ref}-T}{z}}$$
where $\delta_{T_{ref}}$ is the δ value at $T_{ref}$, a chosen reference temperature, and z a microbial-dependent parameter representing the increase in the temperature that leads to a tenfold decrease in the δ value.

The values of $\delta \left(T\right)$ and n under different isothermal conditions were first determined by fitting Equation (5) to the survival curves. The survival parameters involved in Equation (14) were further identified by fitting the equation to the data $\delta \left(T\right)\;vs.\;T$. Then, both path-independent [Equations (7), (13), and (14)] and path-dependent [Equation (9), (13), and (14)] inactivation rate equations were used to predict survival curves under various nonisothermal conditions, and the resulting RMSE values between the measured survival data and their predicted values are listed in Table 2.

As visual examples, the comparisons of results for *B. sporothermodurans* IC4 spores in McIlvaine pH5 under nonisothermal conditions [heating rate (HR) = 1 and 10 °C/min] are illustrated in Figure 3, *Listeria monocytogenes* in ground beef in Figure 4, and *Salmonella senftenberg* in peptone water in Figure 5.

As can be seen from Figure 3, Figure 4 and Figure 5 and the RMSE values listed in Table 2, the path-independent inactivation rate equation provided a more precise prediction of survival curves than the path-dependent one for most of the cases (14 out of 16), experimentally demonstrating that the microbial inactivation rate is most likely independent of path. In addition, the figures showed that the path-dependent inactivation rate equations might overestimate (Figure 3 and Figure 4) or underestimate (Figure 5) the microorganisms’ survival under nonisothermal conditions.

Based on the above discussion, an efficient way to develop a predictive microbial model is to start with an assumption that the inactivation rate does not depend on path, so the rate equation can be uniquely determined from the algebraic primary model under constant conditions [Equation (2)]. If the resulting inactivation rate equation shows unsatisfactory capability to predict survival curves, a trial-and-error process must be used to identify a suitable path-dependent rate equation. Since there are infinite possibilities, it is impossible to find the most suitable or true inactivation rate equation in this case. A logical and effective procedure to construct the right inactivation rate equation from microbial survival data under constant conditions is illustrated in Figure 6. As mentioned earlier, an algebraic function of time that can sufficiently describe the pattern of survival curves under constant conditions can usually be obtained without much difficulty. Using it as a basis, the mathematical structure of an inactivation rate equation can be constructed with a clear purpose, i.e., its integration over time under a constant condition should result in the algebraic function obtained. However, an inactivation rate equation can also be developed directly, as shown in some previous studies [12,13,14,15,16], although this method presents challenges for the reasons mentioned in the introduction section. As for model parameter estimation, a good practice would be refining model parameters estimated from survival data under constant conditions by simultaneously fitting the inactivation rate equation, with each of model parameters being replaced by its secondary model, to multiple survival curves measured under nonconstant conditions. The reason is a perfect constant condition cannot be reached due to the existence of come-up time (CUT), i.e., the time for a constant condition to be reached in an experiment. Therefore, parameters identified from survival data under constant conditions with CUT might have relatively large errors in some cases if CUT was not handled properly [24].

It is worth noting that several studies showed that pretreating a microorganism at a sub-lethal temperature for a sufficiently long time could increase its thermal resistance [60,61]. For the inactivation of a target microorganism in a specific food medium, the extent of the resistance increase depends on the temperature history experienced by the microorganism. In practice, food pasteurization/sterilization processes do not usually have such a phase that can cause changes in microbial resistance to lethal agents used. However, temperature histories experienced by the same raw food material before processing may influence the initial property of the target microorganism, resulting in different resistances of the microorganism to lethal agents used during food processing. This is a scenario different from the path-dependency of the microbial inactivation rate during food processing, which was investigated in the present study.

## 4. Conclusions

It is challenging to come up with a microbial inactivation rate equation directly that can capture accurately the inactivation mechanism of a microorganism when exposed to one or more lethal agents. This study demonstrated that the inactivation rate equation could be constructed from an algebraic primary model established from survival data measured under constant conditions. The inactivation rate of a microorganism may or may not depend on the path defined. The results showed that there was only one rate equation that could be derived from an algebraic primary model if the rate was independent of path. However, if the rate depended on path, there was an infinite number of possible rate equations that all resulted in the same algebraic primary model under constant conditions but, because each of them depended on path in a different way, they made different predictions of survival of the microorganism under the same nonconstant conditions. Based on numerous previous studies as well as the reanalysis of some published microbial survival data, one should first examine the path-independent inactivation rate equation when developing a predictive microbial model. Although not often, if this rate equation does not provide a satisfactory prediction of microbial survival under nonconstant conditions, then path-dependent inactivation rate equations should be constructed and used to predict microbial survival under nonconstant conditions until a suitable one is found.

## Figures and Tables

**Figure 1 foods-14-01980-f001:**
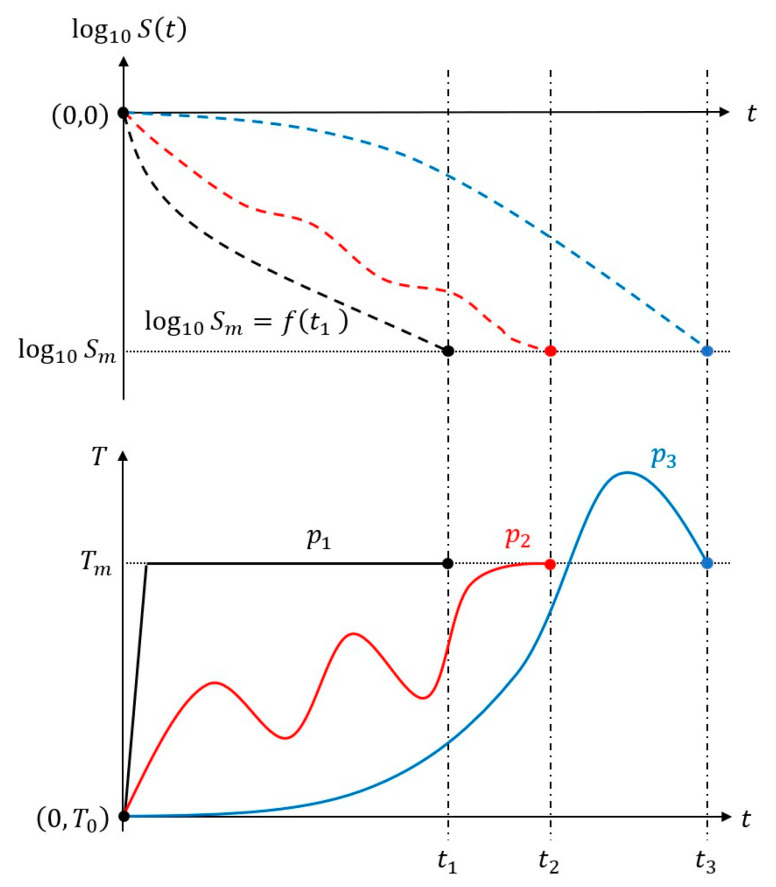
Different temperature profiles or paths to reach a given momentary state $\left[{log}_{10}S_{m},\;T_{m}\right]$ during thermal processing of foods. Solid lines denote paths p1 (black), p2 (red), and p3 (blue), and dashed lines show their corresponding survival curves.

**Figure 2 foods-14-01980-f002:**
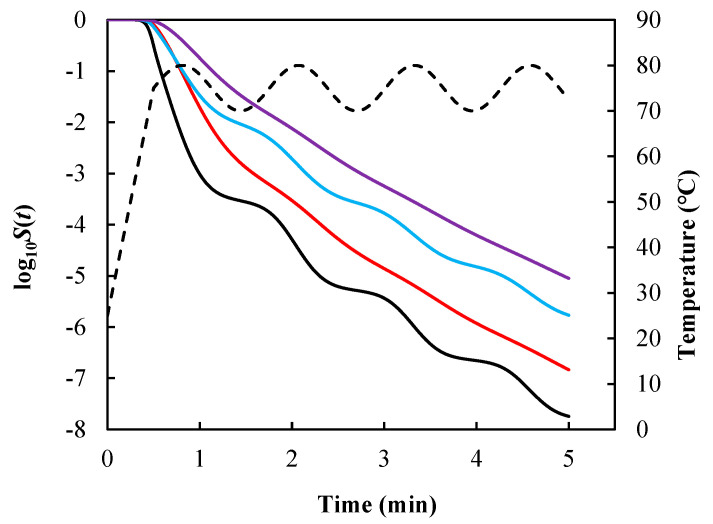
Survival curves of *Salmonella* enteritidis generated using Equation (7) (black line), Equation (8) with m=1 (red line), Equation (9) (blue line), and Equation (10) with m=1 (purple line) under the same temperature condition described by Equation (11) and model parameters k = 0.348 °C^−1^, $T_{c}$ = 66.34 °C, and n = 0.545. Dashed line denotes the temperature profile.

**Figure 3 foods-14-01980-f003:**
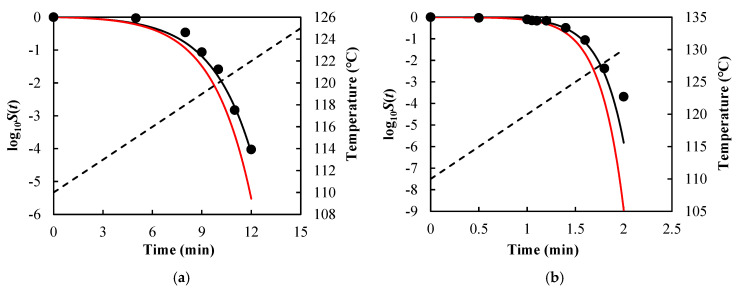
Survival curves of *Bacillus sporothermo-durans* IC4 spores in McIlvaine pH5 predicted using Equations (7), (13), and (14) (black solid lines) and Equations (9), (13), and (14) (red solid lines) with model parameters z = 7.20 °C, $\delta_{T_{ref}}$ = 1.36 min, and n = 1.25 under (**a**) heating rate = 1 °C/min and (**b**) heating rate = 10 °C/min. Filled cycles are experimental survival data [32], and the dashed lines are temperature profiles.

**Figure 4 foods-14-01980-f004:**
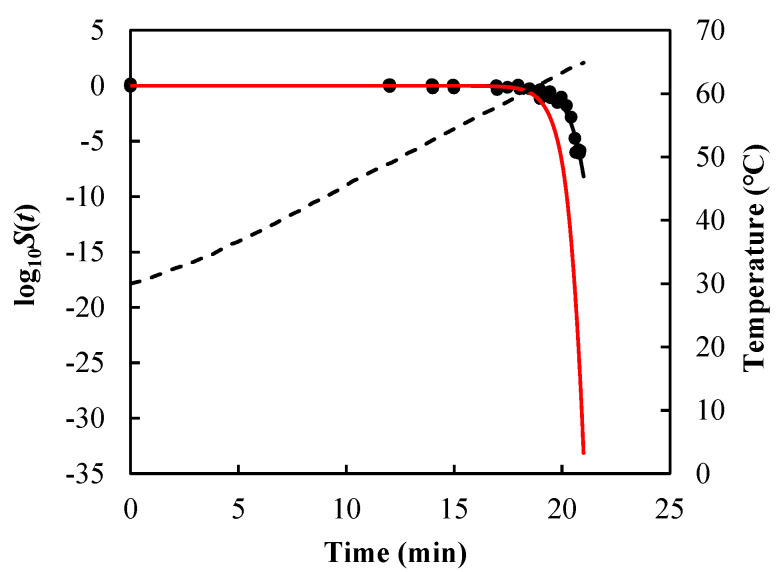
Survival curves of *Listeria monocytogenes* in ground beef predicted using Equations (7), (13), and (14) (black solid line) and Equations (9), (13), and (14) (red solid line) under a nonisothermal condition. Filled cycles are experimental survival data [33], and the dashed line is the temperature profile.

**Figure 5 foods-14-01980-f005:**
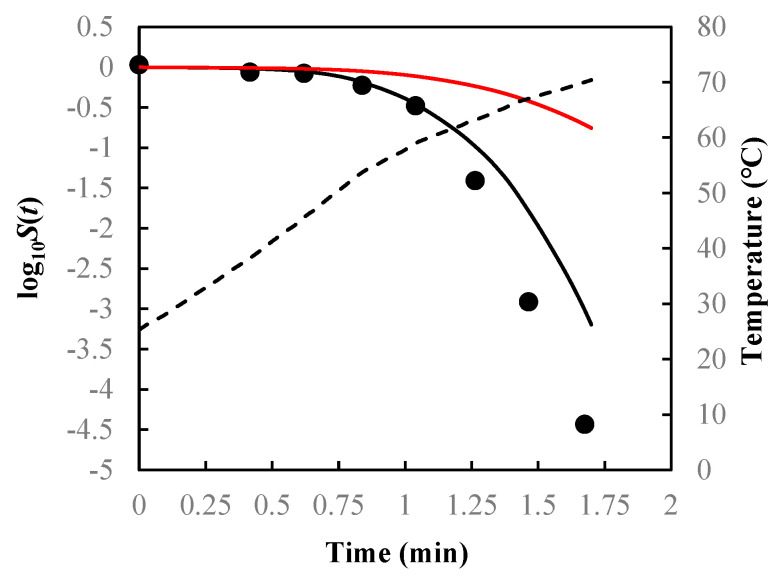
Survival curves of *Salmonella senftenberg* in peptone water predicted using Equations (7), (13), and (14) (black solid lines) and Equations (9), (13), and (14) (red solid lines) under non-isothermal treatment, which were programmed to simulate the heating profile obtained in the heat exchanger with a product flow of 700 mL/min. Filled cycles are experimental survival data [59], and the dashed line is the temperature profile.

**Figure 6 foods-14-01980-f006:**
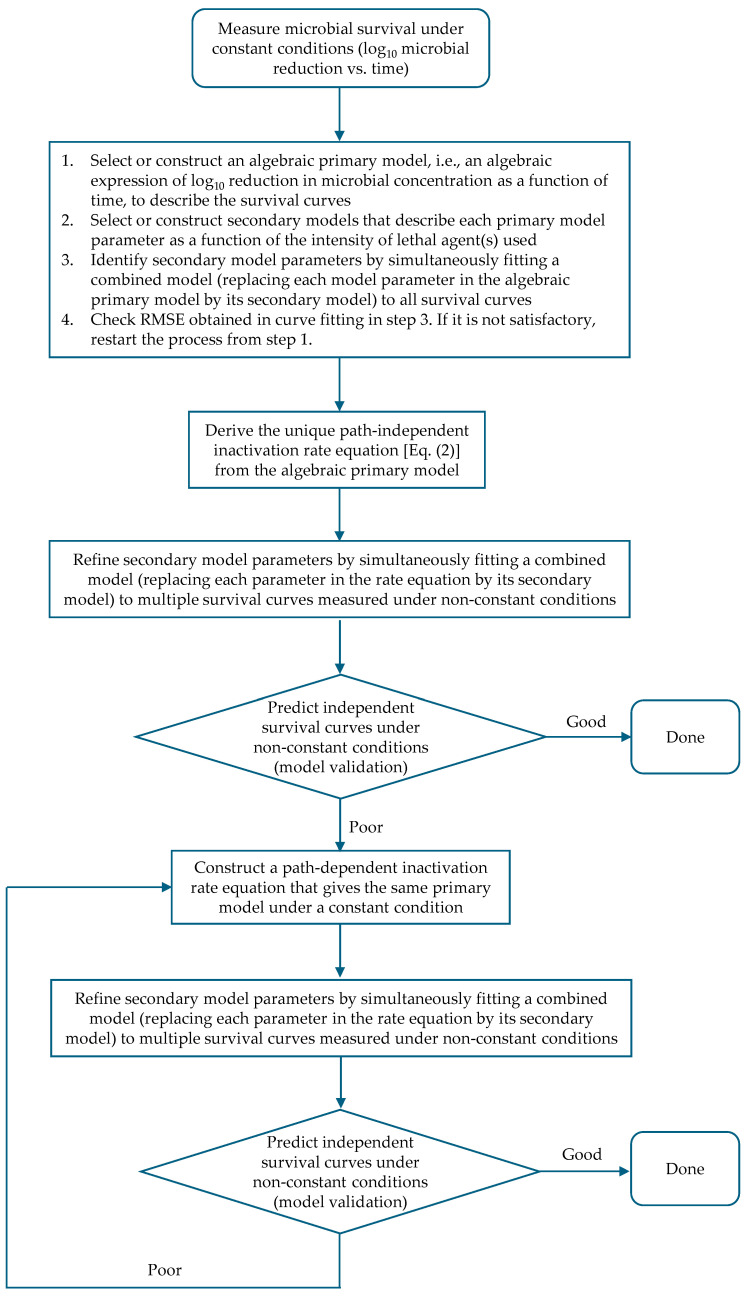
A logical procedure for constructing the microbial inactivation rate equation based on survival data under constant conditions.

**Table 1 foods-14-01980-t001:** The microbial inactivation rate equations, which were developed directly, and the corresponding survival models under constant processing conditions.

Reference	Inactivation Rate Equation	Survival Model Under Constant Processing Conditions
Chick [15]	$\frac{dN\left(t\right)}{dt}=-k\left[L\left(t\right)\right]\cdot N\left(t\right)\;\;\;\;$	$\ln\frac{N\left(t\right)}{N_{0}}=-k\left(L\right)t$
Buckow, Isbarn, Knorr, Heinz and Lehmacher [16]	$\frac{dN\left(t\right)}{dt}=-k\left[L\left(t\right)\right]\cdot {\left[N\left(t\right)\right]}^{n}$	$\ln\frac{N\left(t\right)}{N_{0}}=\frac{1}{1-n}\ln\left\{1+\left[k\left(L\right){N_{0}}^{n-1}t\left(n-1\right)\right]\right\}$
Zhu and Chen [54]	$\frac{dN\left(t\right)}{dt}=-k_{max}\left[L\left(t\right)\right]$ $\;\;\;\cdot \frac{1}{1+C_{C}\left(0\right)\cdot exp\left\{-\int_{0}^{t}k_{max}\left[L\left(t\right)\right]dt\right\}}$ $\cdot \left[N\left(t\right)-N_{res}\right]$	$\log\frac{N\left(t\right)}{N_{0}}=\log\frac{1+C_{C}\left(0\right)+\left[e^{k_{max}\left(L\right)t}-1\right]\frac{N_{res}}{N_{0}}}{C_{C}\left(0\right)+e^{k_{max}\left(L\right)t}}$
Koseki and Yamamoto [14]	$\frac{dN\left(t\right)}{dt}=-k_{max}\left[L\left(t\right)\right]\cdot \left[1-{\left(\frac{N_{min}}{N\left(t\right)}\right)}^{m}\right]\cdot N\left(t\right)\;\;$	$N\left(t\right)={\left[\left(N_{0}^{m}-N_{min}^{m}\right)\cdot e^{-m\cdot k_{max}\left(L\right)\cdot t}+N_{min}^{m}\right]}^{\frac{1}{m}}$

**Table 2 foods-14-01980-t002:** The values of RMSE (unit, log_10_) between the measured survival data and their predicted values using Equations (7) and (9) under various nonisothermal conditions.

Data Source		Bacteria	Food Matrix	Isothermal Treatment	Processing Conditions	Temperature Range	RMSE [Equation (7)]	RMSE [Equation (9)]
Case I [32]	Figure 2(a)	*Bacillus sporothermodurans* IC4 spores	McIlvaine pH7	115 °C, 120 °C, 125 °C, 130 °C	HR = 1 °C/min	110–124 °C	1.45	2.54
	Figure 2(e)				HR = 10 °C/min	110–130 °C	0.382	0.762
	Figure 3(a)				CR = 1 °C/min	110–126 °C	6.82	5.95
	Figure 3(b)				CR = 10 °C/min	110–130 °C	0.437	0.584
	Figure 2(b)		McIlvaine pH5		HR = 1 °C/min	110–124 °C	0.228	0.975
	Figure 2(f)				HR = 10 °C/min	110–130 °C	0.812	2.08
	Figure 2(c)		McIlvaine pH3		HR = 1 °C/min	110–124 °C	0.355	0.447
	Figure 2(g)				HR = 10 °C/min	110–130 °C	1.15	0.851
	Figure 2(d)		Courgette soup		HR = 1 °C/min	110–124 °C	0.395	1.69
	Figure 2(h)				HR = 10 °C/min	110–130 °C	0.836	2.46
Case II [29]	Figure 3(e)	*Geobacillus stearothermophilus* T26	Distilled water	120 °C, 122.5 °C, 125 °C, 127.5 °C	HR = 1 °C/min	90–130 °C	1.41	1.69
	Figure 3(f)				HR = 20 °C/min		0.874	1.12
	Figure 4(c)				With heating, holding and cooling	83–122.5 °C	1.43	1.62
Case III [33]	Figure 6	*Listeria monocytogenes*	Ground beef	57 °C, 60 °C, 63 °C, 66 °C	HR = 1.72 °C/min	30–65 °C	0.389	7.56
Case IV [59]	Figure 6(a)	*Staphylococcus aureus*	Peptone water, performed in the thermoresistometer Mastia	55 °C, 57.5 °C, 60 °C, 62.5 °C	Programmed to simulate the heating profile obtained in the heat exchanger with a product flow of 700 mL/min.	30–65 °C	0.914	1.80
	Figure 6(b)	*Salmonella senftenberg*				30–65 °C	0.832	2.08

## Data Availability

The original contributions presented in the study are included in the article, further inquiries can be directed to the corresponding author.

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
