# Peer review of "Derivation of the Microbial Inactivation Rate Equation from an Algebraic Primary Survival Model Under Constant Conditions"

_foods, 2025, doi:10.3390/foods14111980_

Round 1
Reviewer 1 Report
Comments and Suggestions for Authors
- The article describes an equation under a specific state. Is it universal?
- Does this inactivation rate equation apply to fungi? It is recommended to add relevant examples.
- Is it consistent in thermal and non-thermal sterilization? This should be addressed in the Discussion.
- The first letter of a sentence should be capitalized, such as in lines 100, 113 and 133.
- The meanings of the different colored lines in Figure 2 should be clearly indicated.
- Line 212: "Salmonella Typhimurium" should be changed to "Salmonella typhimurium", and italicized.
- The limitations of this study are not quite reasonable, as mentioning the purpose is only to lay the groundwork for future work.
- Please make sure the conclusions' section underscores the scientific value-added of the paper, and/or the applicability of the findings/results. Highlight the novelty of this study.
- Update the latest references in 5 years.
The English could be improved to more clearly express the research.
Reviewer 2 Report
Comments and Suggestions for Authors
The the applicable temperature range, microbial types, microbial metabolic characteristics (such as metabolic characteristics, heat resistance), and food medium properties of the derived inactivation rate equation should be discussed.
3.2. A revisit to previous studies. In addition to case studies, there should also be some experimental verification.
Reviewer 3 Report
Comments and Suggestions for Authors
Thank you for the opportunity to review this manuscript. I appreciate the effort of the authors to try to give more theoretical meaning to inactivation models. However, I have several concerns about the relevance, clarity, and scientific justification of the proposed methodology.
My primary concern lies in the central premise of the study: the focus on using what the authors call primary models, like Weibull, for microbial inactivation, even when there are temperature changes (for the sake of clarity I will use as example temperature in my justifications, but other factors can be considered). While I understand that such an approach might have been more relevant in the past (due to software and hardware limitations), current modelling approaches allow for much more flexible and accurate alternatives. Today, it is straightforward to work with inactivation rates (derivative forms) and incorporate secondary models directly. The resulting system of ordinary differential equations (ODEs) can be efficiently solved using modern numerical solvers, available in most programming languages, without imposing restrictive assumptions on temperature or the form of the inactivation rate. This also enables the modelling of more complex scenarios in a very fast way, such as spatial heterogeneity in temperature, through partial differential equations, which have even been employed in the context of real-time control and optimization in food systems.
I would respectfully disagree with the notion presented in the manuscript that there is no consensus about which method is more appropriate: the method in [15] or the alternatives in the literature (line 56). On the contrary, recent literature trends indicate a clear preference for numerically solving the full set of differential equations, as supported by the publication dates and methodologies of the cited references. Additionally, the statements about the alternative methods are incorrect: they do not differentiate both sides of the primary model equation under constant conditions; instead, they start from the inactivation rate and use the integrative form only when conditions are constant—a very important difference and a recurring issue throughout the work.
Nevertheless, recognizing that the approach used here may still be of interest in certain contexts, I focus my comments below mostly on the specific presentation and arguments in the work.
- Title Clarity: The current title, “Derivation of the inactivation rate equation from a primary microbial survival model,” suggests a broader or different scope than what is actually investigated. The paper primarily explores the idea of path dependence in microbial inactivation, which could be better reflected in a revised title.
- Practical Relevance: The manuscript does not clearly explain the practical advantages of using the approach in [15] over the more conventional approach of solving Eq. (2) with time-dependent temperature profiles. In classical modelling temperature affects the inactivation rate, not survival directly. There are multiple examples: kinetic models, mass/energy balance equations, population equations, etc.
- Terminology – “Path Dependence”: The use of the term "path dependent" is potentially confusing, as path functions may have different meanings in modelling contexts. In line 151, the manuscript clarifies that the “path” refers to the temperature profile. However, many sentences become nonsensical when replacing the term with “temperature profile dependence.”
- Model Justification (Eq. 9): The introduction of Eq. (9) is not justified. The assumption regarding the functional dependence of model parameters on C lacks a theoretical or empirical basis. It is just an example as they stated in the work. While the chosen form may conveniently lead to the desired conclusion, it may not hold in more general cases. Note that β=β(C) when C is a constant; however, this relationship does not necessarily hold for other expressions.
- Support for Conclusions:
- The statement that “there exists an infinite number of possible microbial inactivation rate equations that result in one primary survival model under constant conditions” is not demonstrated rigorously. The examples provided illustrate that different temperature profiles provide different survival dynamics (and in Figure 1 the states are the same but at different times) but do not substantiate the general claim.
- “However, only one of them is path-independent and can be derived from the primary model using Eq. (2).” Assuming that path refers to the temperature profile, I do not see that this has been proven in the article, and I also do not agree. Why cannot a temperature profile changing with time be included in Eq. (2) and solved numerically? I have read the referenced work [15], and it does not make this claim. In fact, it explains that this is done when using kinetic models.
- “Since many studies found that the inactivation rate did not depend on path, an efficient and effective way to derive the inactivation rate equation from a primary survival model is to examine the validity of the path-independent one first. If it shows unsatisfactory capability to predict survival curves, a trial-and-error process must be used to identify a path-dependent one; however, it is impossible to find the most suitable or true inactivation rate equation from infinite possibilities.” I am not sure what the authors are trying to say with these sentences. Also I cannot find which papers assume no dependence on temperature profiles which is related with the following key point.
- Misinterpretation of Previous Studies: The claim that In some previous studies, "the inactivation rate equations were obtained by taking the derivative of both sides of a primary model under constant conditions. This method implicitly assumed that the inactivation rate depends on path since only the one derived using Eq. (2) does not." is not true. These studies begin from the inactivation rate equation and use their integrative forms when there is an easy-to-find analytical solution, which usually is the case for constant temperature profiles. Note also that this is the general approach in modelling across many other fields—studying rates of change and their dependencies on temperature and other factors, and providing analytical forms only when analyitical integration is possible.
Recommendation:
Due to the concerns outlined above—particularly regarding the scientific justification, clarity, and practical relevance of the proposed approach—I do not believe the manuscript is currently suitable for publication in Foods in its present form.
